# Risk-Stratified Breast Cancer Screening Incorporating a Polygenic Risk Score: A Survey of UK General Practitioners’ Knowledge and Attitudes

**DOI:** 10.3390/genes14030732

**Published:** 2023-03-16

**Authors:** Aya Ayoub, Julie Lapointe, Hermann Nabi, Nora Pashayan

**Affiliations:** 1National Heart and Lung Institute, Imperial College London, London SW3 6LY, UK; 2Oncology Division, CHU de Québec-Université Laval Research Center, Québec City, QC G1R 3S3, Canada; 3Department of Social and Preventive Medicine, Faculty of Medicine, Université Laval, Québec City, QC G1V 0A6, Canada; 4Department of Applied Health Research, University College London (UCL), London WC1E 7HB, UK

**Keywords:** risk-stratified screening, polygenic risk score, breast cancer, GP survey

## Abstract

A polygenic risk score (PRS) quantifies the aggregated effects of common genetic variants in an individual. A ‘personalised breast cancer risk assessment’ combines PRS with other genetic and nongenetic risk factors to offer risk-stratified screening and interventions. Large-scale studies are evaluating the clinical utility and feasibility of implementing risk-stratified screening; however, General Practitioners’ (GPs) views remain largely unknown. This study aimed to explore GPs’: (i) knowledge of risk-stratified screening; (ii) attitudes towards risk-stratified screening; and (iii) preferences for continuing professional development. A cross-sectional online survey of UK GPs was conducted between July–August 2022. The survey was distributed by the Royal College of General Practitioners and via other mailing lists and social media. In total, 109 GPs completed the survey; 49% were not familiar with the concept of PRS. Regarding risk-stratified screening pathways, 75% agreed with earlier and more frequent screening for women at high risk, 43% neither agreed nor disagreed with later and less screening for women at lower-than-average risk, and 55% disagreed with completely removing screening for women at much lower risk. In total, 81% felt positive about the potential impact of risk-stratified screening towards patients and 62% felt positive about the potential impact on their practice. GPs selected training of healthcare professionals as the priority for future risk-stratified screening implementation, preferring online formats for learning. The results suggest limited knowledge of PRS and risk-stratified screening amongst GPs. Training—preferably using online learning formats—was identified as the top priority for future implementation. GPs felt positive about the potential impact of risk-stratified screening; however, there was hesitance and disagreement towards a low-risk screening pathway.

## 1. Introduction

Breast cancer (BC) is the most commonly diagnosed cancer in the United Kingdom (UK) and the leading cause of premature death in women aged 30–60 years old [1]. The National Health Service Breast Screening Programme (NHSBSP) offers mammographic screening to women 50–70 years old every 3 years. However, this ‘one-size-fits-all’ approach does not reflect the wide range of individual risks amongst women [2]. A UK independent panel review found that although there is a 20% relative reduction in BC death following screening, 19% of screen-detected cancers are likely to be overdiagnosed [3].

A polygenic risk score (PRS) captures the aggregated effect of many common low-impact genetic variants—single nucleotide polymorphisms (SNPs)—derived from genome wide association studies (GWAS). A ‘personalised breast cancer risk assessment’ (PBCRA) combines PRS with other genetic and nongenetic risk factors to stratify women into risk groups and then tailor screening to each risk group by varying the start/stop age, frequency, and modality of screening. Risk-stratified screening for breast cancer promises to improve the cost-effectiveness and the benefit–harm balance of population-based breast screening programmes [2]. Risk stratification may lead to earlier diagnosis and improved patient outcomes for women at high risk of breast cancer through targeted mammographic surveillance and triggering more timely advice on lifestyle and chemoprevention. Risk stratification may also potentially reduce harms from overdiagnosis and false positive findings, through identification of low-risk women requiring less mammographic surveillance [4,5,6]. PRS testing of selected SNPs using DNA microarrays is quick and low-cost.

PRSs are derived from GWAS summary statistics and are associated with BC subtypes estrogen receptor (ER) positive and ER negative [7]. A PRS of 313 SNPs—‘PRS^313^′—has been consistently associated with risk of developing BC, accounting for approximately 20% of the polygenic risk of BC [8]. However, PRS^313^ is developed based on data from European ancestry women [8]. Women in the top 1% of PRS^313^ distribution have 4.4- and 2.8-fold risks for ER-positive and ER-negative BC, respectively—these are risk levels close to or on a par with pathogenic variants in, e.g., *BRCA1/2* [8]. Women in the lowest 1% of PRS^313^ risk are also at significantly low risk of developing ER-positive and ER-negative BC—0.16- and 0.27-fold risks, respectively [8]. Further, in a study comparing the population-level predictive ability of PRS, family history, and pathogenic variants, PRS was found to be the most predictive for identifying women in the general population at high risk of developing BC, and only PRS identified any women at low risk [9]. Adding pathogenic variants to population stratification models has limited predictive value, owing to their rarity in the population, whereas PRS is an effective contributor to population risk stratification [10]. There are several large international studies underway evaluating the impact of PRS implementation as part of a risk-stratified BC screening programme: ‘MyPeBS’ (Europe) [11], WISDOM (United States) [12], PROCAS (UK) [13], PEREPECTIVE I&I (Canada) [14].

Understanding healthcare professionals’ (HCPs’) knowledge and attitudes is key to evaluating the acceptability of risk-stratified BC screening. To date, studies have scoped views of HCPs working in secondary care and genetics services currently using PRS on a research basis; they were small focus groups [13,15,16,17] and qualitative analysis of surveys [18]. A quantitative survey of HCPs (*n* = 593) working in Canada has recently been published [19].

In the UK, breast screening is provided by the NHSBSP. Nevertheless, as the first contact for patients, GPs receive queries from patients related to risk information, screening recommendation, and about interventions to modify risk level. Thus, GPs will need to be knowledgeable in genomic concepts underpinning screening and confident interpreting results and explaining the risks and benefits of screening to support patients with informed decision making [18,20,21]. GPs deliver the bulk of patient care in the NHS and play a pivotal role coordinating a patient’s healthcare across primary and secondary care services. Effective implementation of a future risk-stratified NHSBSP will depend on the understanding and acceptability of risk-stratified screening by GPs. Many countries around the world, including those trialling risk-stratified BC screening using PRS—‘MyPeBS’ (Europe) [11], WISDOM (United States) [12], PROCAS (UK) [13], and PEREPECTIVE I&I (Canada) [14]—also adopt a similar structure to their healthcare system, with family doctors acting as the mainstay of healthcare interactions; thus, this study holds international relevance.

To our knowledge, this study is the first to survey GPs in the UK with the aims of exploring their: (i) knowledge of risk-stratified screening—covering self-reported familiarity with PRS and confidence communicating concepts underpinning testing; (ii) attitudes towards risk-stratified screening; (iii) preferences for continuing professional development. As primary care providers play a pivotal role in the NHS in the UK and in international healthcare systems, understanding their knowledge, attitudes, and learning preferences will be an important step towards evaluating the acceptability of future risk-stratified BC screening.

## 2. Materials and Methods

### 2.1. Survey Population and Administration

A cross-sectional study using an online anonymous survey was conducted between July and August 2022 (Appendix A). Following pilot testing, the survey was distributed via the Royal College of General Practitioners to members who accepted receiving research invitations, 17 UK GP groups on social media and local and regional GP mailing lists. The distribution strategy and survey questions were aimed at capturing results from UK GPs not necessarily specialised in genetics. A minimum of 96 participating GPs were required to achieve a 10% margin of error at a 95% confidence interval.

### 2.2. Survey Design

Background information about PRS and risk-stratified screening was provided in the survey introduction and within specific questions. The survey consisted of 18 questions. The first five questions collected information about the participants’ professional and sociodemographic characteristics, and the remaining questions covered the three key themes (Box 1).

Questions were designed following a review of past studies covering HCPs’ knowledge and attitudes towards risk-stratified screening [13,15,16,17,18,19,22]; to facilitate comparison with the only other quantitative survey of HCPs (in Canada), six questions were borrowed directly or adapted from this survey [19].

Box 1Outline of the survey.
**Professional and sociodemographic characteristics (5 MCQ questions)**
GP roleYears of experienceLocation of practiceGenderEthnicity
**Knowledge of PRS and risk-stratified screening incorporating PRS (4 sliding scale questions)**
Familiarity with the concept of PRSConfidence communicating to a patient:
○polygenic inheritance○advantages and disadvantages of a personalised breast cancer risk assessment (PBCRA)○PBCRA result as a 10-year absolute risk

**Attitudes towards future implementation of risk-stratified screening (6 questions)**
Views on the current National Health Service Breast Screening Programme (NHSBSP) (Likert)Views on different targeted screening approaches (Likert)Perception of potential impact of risk-stratified screening (sliding scale):
○on both patients○on general practice
Views on aspects of the NHS that would need to be enhanced to deliver risk-stratified screening (multiple choice question, MCQ)

**Preferences for continued professional development (2 MCQ questions)**
Desired clinical resourcesFormats for learning more about risk-stratified screening
**At the end of the survey (2 free-text questions)**
How they received the surveyAny additional comments

### 2.3. Statistical Analysis

All survey responses were reviewed to assess the level of completeness. The dataset was extracted within Qualtrics. All five GP characteristics, sliding-scale, and Likert-scale responses were bucketed to improve the confidence of statistical analysis (Appendix A).

Descriptive statistics were used to summarise participant responses. Chi-squared tests explored whether respondents’ answers to questions about knowledge and attitudes differed according to their professional and sociodemographic characteristics. Fisher’s exact test was used when predicted cell size was <5. All statistical analyses were performed using Qualtrics [23]. Bonferroni correction was used to adjust the *p*-value to correct for the multiple comparisons problem and, therefore, reduce the chance of a type I error. After Bonferroni correction, *p* < 0.003 was used for statistical significance. At 5% significance level, there would have been a statistically significant association between GP role and preference for risk-stratified screening. Free-text responses were analysed thematically using key topic labels using Qualtrics.

## 3. Results

### 3.1. Characteristics of Survey Respondents

A total of 109 GPs completed the survey. The lowest response rate to any question was 84% (*n* = 92). The respondents were: 39% (*n* = 43) salaried GPs; 31% (*n* = 34) GP partners; 13% trainee GPs (*n* = 14); 10% locum GPs (*n* = 11); and 6% selected ‘Other GP’ (*n* = 7) (Appendix A). GPs in all parts of the UK, except Wales, participated; 35% (*n* = 38) of respondents were in London. In total, 57% (*n* = 62) of GPs were female. More than half of the respondents were ethnically White (58% (*n* = 63)); the second-largest ethnic group was Asian or Asian British (25% (*n* = 27)), followed by Black (6% (*n* = 6)), ‘Other ethnic group’ (5% (*n* = 5)), and mixed/multiple ethnic groups (3% (*n* = 3)).

### 3.2. Knowledge of PRS and Risk-Stratified Screening Incorporating PRS

In total, 49% (*n* = 45) of participants stated they were not familiar with PRS; 43% (*n* = 38) responded that they were slightly to moderately familiar and 9% (*n* = 9) responded that they were very to extremely familiar (Figure 1). One third of GPs across all three questions on communication rated themselves as ‘not confident’. In total, 23% (*n* = 23) of GPs reported feeling not confident explaining polygenic inheritance and 32% (*n* = 33) reported feeling not confident explaining a PBCRA result to a patient as a 10-year absolute risk. The lowest scoring question was on explaining the advantages and disadvantages of PBCRA, where 43% (*n* = 40) of the GPs reported feeling not confident.

### 3.3. Attitudes towards Future Implementation of Risk-Stratified Screening

Most respondents, 74% (*n* = 80), strongly/somewhat agreed, 11% (*n* = 12) neither agreed nor disagreed, and 16% (*n* = 16) strongly/somewhat disagreed that the current UK NHSBSP is an effective method for early detection of BC.

Regarding views on targeted screening based on PBCRA (Figure 2), 80% (*n* = 86) of GPs agreed that women at high risk should have earlier and more frequent screening. Views on more frequent screening for high-risk women were also very supportive, although fewer GPs were in strong agreement with this approach. For women at higher-than-average risk, results were very similar for both questions on either increasing the frequency of screening +/− starting screening earlier: c. 60% of GPs agreed, c. 30% neither agreed nor disagreed, and c. 6% disagreed. For women at lower-than-average risk, results across all three targeted screening strategies (later, less frequent screening, and both later and less screening) were very similar: c. 20% of GPs agreed; c. 40% of GPs neither agreed nor disagreed and 40% disagreed. For women at much lower risk, most GPs did not support removing screening completely: 55% (*n* = 57) of GPs disagreed—half of whom strongly disagreed (*n* = 29)—and only 11% (*n* = 11) agreed.

Regarding GPs’ perception of the potential impact of risk-stratified screening on patients and general practice, 81% (*n* = 86) felt it would have a positive impact on patients; 62% (*n* = 65) felt positive about the impact on general practice.

The top four choices for aspects of the NHS that should be enhanced to implement risk-stratified screening were: training of HCPs (13%); number of genetic counsellors (12%); number of primary care physicians (11%); time allocated to a patient–physician appointment (11%) (Figure 3). The number of geneticists was seen as a lower priority (6%) and no respondents felt that the healthcare system was ready to implement risk-stratified screening.

### 3.4. Preferences for Future Learning about Risk-Stratified Screening

The top 3 choices to better understand risk-stratified screening were: information on the basics of PBCRA (15%); information on the calculation of PRS (14%); and information on interpreting the results of PBCRA (14%) (Figure 4). Respondents favoured online formats for learning, opting for online courses (22%); access to a website (18%); or a webinar-type conference (16%) as their top 3 choices for learning (Figure 5).

At the end of the survey, 25 free-text comments were analysed thematically. The top five concerns raised by GPs were: lack of capacity to take on additional workload; lack of knowledge of PRS; that PBCRA was too ‘complex’ and ‘specialist’ for a GP appointment; the need to evaluate the psychological impact of PBCRA on patients; and the potential for PBCRA to exacerbate health inequity.

There were no statistically significant associations at the 0.003 level found between results assessing knowledge and attitudes towards PRS and PBCRA and the professional and sociodemographic characteristics of respondents ( Appendix A).

## 4. Discussion

### 4.1. Main Findings

The study reveals limited knowledge of PRS and risk-stratified screening amongst GPs, with training in the field—preferably using online learning formats—identified as the top priority for future implementation. GPs felt positive about the potential impact of risk-stratified screening; however, there was hesitance and disagreement towards a low-risk screening pathway.

### 4.2. Strengths and Limitations

GPs across all career stages and roles participated; the breakdown of respondents’ gender and ethnicity closely mirrors GP workforce demographic information held by the General Medical Council [24]. The minimum number of responses needed for a 10% margin of error was surpassed and the findings of this study are consistent with the Canadian survey [19].

Use of a Bonferroni-corrected *p*-value at 0.003 was necessary to adjust for multiple comparisons. The study found no statistically significant relationships between the demographic and career-stage characteristics of the participating GPs and their responses. Otherwise, at 5% significance level we would have seen an association between being a GP trainee and preference for targeted screening approaches. However, interpretation of trainee responses was limited by the sample size (*n* = 14) and they were significantly outnumbered by qualified GPs (*n* = 93) in this survey. Another possible reason for the lack of statistically significant results could be the closeness of percentages in the alternatives for the question items provided. Given these factors, the small sample size made identifying statistically significant differences in responses across sociodemographic and professional variables challenging. A larger number of participants could help tease out potential relationships between the questions posed and GP role and years of experience.

The survey was intended to be brief to maximise participation, with no prior specialist knowledge required. Some respondents, however, felt more information about the performance of PRS needed to be included to make informed responses. Numerical descriptions of the 10-year absolute risk thresholds assigned to different targeted screening approaches were also omitted from the survey; this may have challenged respondents when completing this question, especially since lower-risk thresholds are not used in current practice. The survey was only online and this may have biased responses for preferred training modality. Further, an additional question enquiring about participants’ prior experience working in genetics may have highlighted selection bias.

### 4.3. Comparison with Existing Literature

#### 4.3.1. Familiarity with PRS and Confidence Communicating PBCRA Is Low amongst GPs

The results of this study show that familiarity with PRS was limited amongst UK GPs in our sample across different professional roles and years of experience; just over half of respondents (51%) were familiar with PRS, with most of these respondents only ‘slightly familiar’. Limited familiarity with PRS amongst GPs is echoed by qualitative studies of HCPs working in genetics [15,18], and in the quantitative study of HCPs in Canada, where 72% of respondents reported being ‘very unfamiliar’ or ‘not knowing’ the concept of PRS at all [19].

While it is more likely that a future risk-stratified NHSBSP will continue to be delivered as a stand-alone programme [13], UK GPs will need to know basic genomic concepts, be confident interpreting results and explain the risks and benefits of PBCRA to support patients with informed decision making [18,20]. Questions in this study are focussed on basic knowledge and competencies that may be expected if implementation followed pathways and reporting strategies being trialled. Patients and GPs will possibly not receive separate polygenic risk score (PRS) reports, but the risk category would probably be returned directly to the patient by post, similar to current screening programmes, conveying a 10-year absolute risk and recommended risk-tailored interventions. Patients at high (≥8%) or moderate (5–7.9%) risk would be offered an appointment to discuss more frequent screening and chemoprevention, as per NICE guidance [25]. In the ongoing UK PROCAS trial, follow-up is conducted in specialist clinics, with trained counsellors and clinicians; however, it is not clear how PBCRA reporting and follow-up could be conducted on a population level [20].

GPs will likely play a role counselling and following-up patients receiving PBCRA [20,26]; studies have also shown a preference by patients for receiving follow-up in this setting [18,27]. However, this study suggests most GPs do not feel confident taking on this task, which is perhaps not surprising, given documented low levels of confidence and knowledge in genetics amongst the primary care workforce [28,29], combined with the relative novelty of PRS and PBCRA. Until recently, the field of genetics has focused on rare monogenic conditions not commonly encountered in general practice.

Studies evaluating how HCPs working in genetics convey PRS and PBCRA results also highlight unfamiliarity with these concepts and a tendency to adopt a more clinician-centred, biomedical focus to counselling, in contrast to the rapport and counselling techniques usually employed when discussing monogenic conditions [30,31,32]. GPs in this study felt slightly more confident communicating polygenic inheritance, but less confident conveying the meaning of a result or the advantages and disadvantages of testing. Since GPs may take on more of a counselling role in a future risk-stratified NHSBSP, the development of clinical guidelines is required [16] to ensure risk is conveyed clearly to patients. Indeed, training of HCPs was highlighted as the top NHS priority by survey respondents to enhance future implementation of risk-stratified screening.

As the availability of genomic testing rapidly grows, easily accessible targeted interventions to improve competencies will be needed, particularly amongst the primary care workforce [33]. The preference for online learning to learn more about PBCRA amongst respondents is consistent with results of previous studies [18,19,34]. Online courses and websites can be accessed at the point of need; this preference may reflect the competing priorities on GPs’ time in the busy clinic. Across several survey questions, GPs emphasised limited time and capacity as barriers to delivering consultations around PBCRA.

#### 4.3.2. GPs Support Risk-Stratified Screening but Have Reservations about a Low-Risk Screening Pathway

This study demonstrates that GPs support the principle of a risk-stratified NHSBSP but mainly view the benefits of this as a means of enhancing screening for women at high and higher-than-average risk of BC. Further, whilst the majority of GPs did agree that the NHSBSP is effective at detecting BC early, only 2 in 10 GPs strongly agreed, indicating a desire for a more effective screening strategy for early BC detection.

Limited knowledge of PRS may offer one explanation for the large proportion of ambivalent survey responses to support later or less frequent screening for women at low risk. GPs selected ‘information on the basics of PBCRA’, ‘calculation of PRS’, and ‘interpreting PBCRA results’ as their top three priorities for continued learning to better understand risk-stratified screening. An average of 42% of GPs across the three questions related to screening proposals in low-risk groups neither agreed nor disagreed with the proposal. However, over half of GPs disagreed with removing screening completely for patients at much lower risk. Free-text comments later in the survey also revealed GPs wished to learn more about the evidence base for proposals, feeling unequipped to answer questions about targeted screening without this information. In contrast, attitudes towards increased mammographic surveillance for women at higher risk were very supportive; this may reflect familiarity with current evidence-based guidelines which support earlier and more frequent mammographic screening for women at higher risk [35].

Hesitancy and unease towards implementing low-risk screening pathways has also been highlighted in recent UK focus group studies of HCPs trialling risk-stratified screening [13,17]. In these studies, HCPs raised fears around patients interpreting low risk as no risk [17]; concerns around the stability of 10-year risk estimates [17]; and fears of possible blame or litigation in case of an interval cancer developing [13].

Studies have also found hesitance amongst women to accept reduced surveillance, since they found regular screening reassuring [36,37,38]. Women integrate PBCRA with their own pre-existing health beliefs and notions of risk [39]. A systematic review found that low-risk PBCRA results did not necessarily reduce ‘perceived susceptibility’ amongst women [40], echoing concerns from GPs in the survey regarding public perception of removing screening.

#### 4.3.3. Risk-Stratified Screening Requires Evaluation of Potential Psychological Harms to Women

Whilst GPs were generally supportive of risk-stratified screening, a point of worry was the need to evaluate the psychological impact of risk-stratified screening on women. GPs raised concern that PBCRA might result in false reassurance or undue anxiety.

Large-scale studies are assessing the psychological impact of PBCRA on patients [14,25] which, to date, is not well understood [25]. Research has shown there is public interest in a risk-stratified approach [38,41]. However, studies have highlighted the complexity of attitudes and emotional responses amongst women receiving test results, including differences in attitudes according to ethnicity [37,42].

The reporting strategy is an important determinant in how patients react to and process their results [26]. Positive attitudes towards PBCRA relate to the value of results to inform decision making and deliver ‘result actionability’ [36]. In the majority of studies, PBCRA results are well-received by women at high risk [30,36,37,39,43,44,45,46], although one prospective study reported greater ‘genetic testing-specific distress’ and ‘decisional regret’ in women with a high PRS than in those with a low score [47].

Some GPs in this survey used phrases such as ‘opening pandora’s box’ and ‘dropping a hand-grenade’ to describe screening using PRS. A recent systematic review found that women only felt positive about risk-stratified screening if results did not just depend on receiving genetic information, but also increased knowledge, engendered a sense of empowerment, and supported women to make healthy lifestyle changes [40], essentially equipping women with the means to change their risk.

Building workforce capacity and capabilities will be needed to support greater numbers of women identified at higher risk. Combining PRS into risk models enables identification of a larger proportion of women at high risk of BC who would benefit from risk-reducing chemoprevention [48,49]. The PROCAS study calculated a PBCRA for 54,000 women in England, using a PRS of 18 SNPs; the study demonstrated population risk stratification could identify a further 2.5% of women in England at high risk and an additional 10% at moderate risk [50].

Whilst breast screening is delivered independently of general practice, GPs play a crucial role in the NHS, coordinating a patient’s care and delivering 300 million annual patient consultations [51]. GPs would also prescribe and monitor side effects from chemoprevention and offer lifestyle advice and management. This study highlights the important role of GPs in population health. Specialised training in communication of PRS and PBCRA will be a crucial factor in mitigating psychological harms to patients [15]. Results of the survey emphasise the need for training and stronger capacity in primary care—increased staffing and time—as key priorities to be enhanced for any future implementation of risk-stratified screening.

#### 4.3.4. More Research Is Needed to Evaluate the Effects of Risk Stratification on Health Inequity

GPs in the survey raised concerns that using a test validated by White European women only would exacerbate existing health inequity. Whilst evidence is growing in favour of risk-based screening, there is wide recognition that until PRS is validated across different ethnicities, this poses a major barrier to clinical implementation [52,53]. Studies of attitudes towards PRS amongst genetic counsellors found that they are reluctant to offer PRS in practice, when it can only be used in women of White European ancestry [13,31,32]. PRSs have been shown to be highly reproducible across European cohorts [54,55], but studies examining PRS models across different ethnicities have shown smaller effect sizes, particularly amongst women of African ancestry [54,56,57,58,59]. Whilst large-scale research programmes like ‘Our Future Health’ [60] in the UK and CONFLUENCE in the US [61] aim to increase ethnic diversity in genomic datasets through deliberate sampling of ethnic minority groups, the research gap remains large [62].

In the survey, GPs not only highlighted concerns around ethnic disparities, but also that other socioeconomic barriers will need evaluation to ensure equitable implementation of risk-stratified screening. Currently, minority ethnic groups, as well as those from a low socioeconomic background, are less likely to attend screening [10,63]. Whilst BC incidence and mortality rates are lower in Black and Asian minority ethnic groups compared to White females in England [64], this may reflect unmet need [13]. Further, 3000 BC cases in England are linked to lower deprivation each year and BC mortality rates in England are higher amongst females living in the most deprived areas [64]. Existing low-levels of engagement with NHSBSP are shown to be linked to a range of complex factors—culture, low educational attainment, and language barriers [10,13,63]. Modelling for risk stratification requires patients to complete an online self-questionnaire [25]. However, lower health literacy is known to be associated with reduced uptake of preventive services [65]. GPs in this study have mirrored concerns that barriers to access may be amplified through adopting risk stratification [17,54], which presents more complexity than the current NHSBSP.

### 4.4. Implications for Research and Practice

Equipping the NHS workforce with the capacity and capabilities to engage with advances in genomic medicine, covering an appreciation of best practices and referral as well as an understanding of ELSI, which received an intermediate rating in our study, is a current NHS priority. Consequently, the UK government has established the Genomics Education Programme (GEP) [66].

Risk stratification allows better deployment of resources to where they are needed most and has the potential to improve the cost-effectiveness and benefit–harm balance of screening. Whilst the use of a PRS within a population risk-stratification programme is in development, it is available through direct-to-consumer tests for a variety of common, complex conditions as well as through the ‘CanRisk’ tool in the NHS [67]. This situation highlights a need to improve awareness of the field through the creation of resources at the point-of-need to support GPs with clinical decision making.

Further, specialised training in communication of polygenic risk and PBCRA will be important to mitigate psychological harms to patients [15]. GPs, who are likely to adopt more of a counselling role, will need to communicate both the implications and limitations of testing, presenting balanced information to women about their harms and benefits [19]. Indeed, there are no studies examining the impact of PBCRA communication by HCPs who are not already working in specialist BC or genetics clinics; this will be required to understand knowledge gaps that should be reflected in curriculum design and referral pathways.

## 5. Conclusions

This study provides the first insight into the knowledge and attitudes of GPs towards risk-stratified breast cancer screening incorporating a polygenic risk score. The exploratory results reveal limited knowledge of PRS and risk-stratified screening amongst GPs, with training in the field—preferably using online learning formats—identified as the top priority for future implementation. GPs felt positive about the potential impact of risk-stratified screening; however, there was hesitance and disagreement towards a low-risk screening pathway.

Whilst UK GPs may not be directly involved in the delivery of a future National Health Service Breast Screening Programme, GPs deliver the bulk of patient consultations in the NHS, are the first point of contact for patients, coordinate patient care, and would manage lifestyle and chemopreventive measures for high-risk women in the community. Effective implementation of a future NHSBSP will depend heavily on education and training efforts to enhance genomic competencies amongst the primary care workforce and the acceptability of a screening pathway for women at low risk for breast cancer.

## Figures and Tables

**Figure 1 genes-14-00732-f001:**
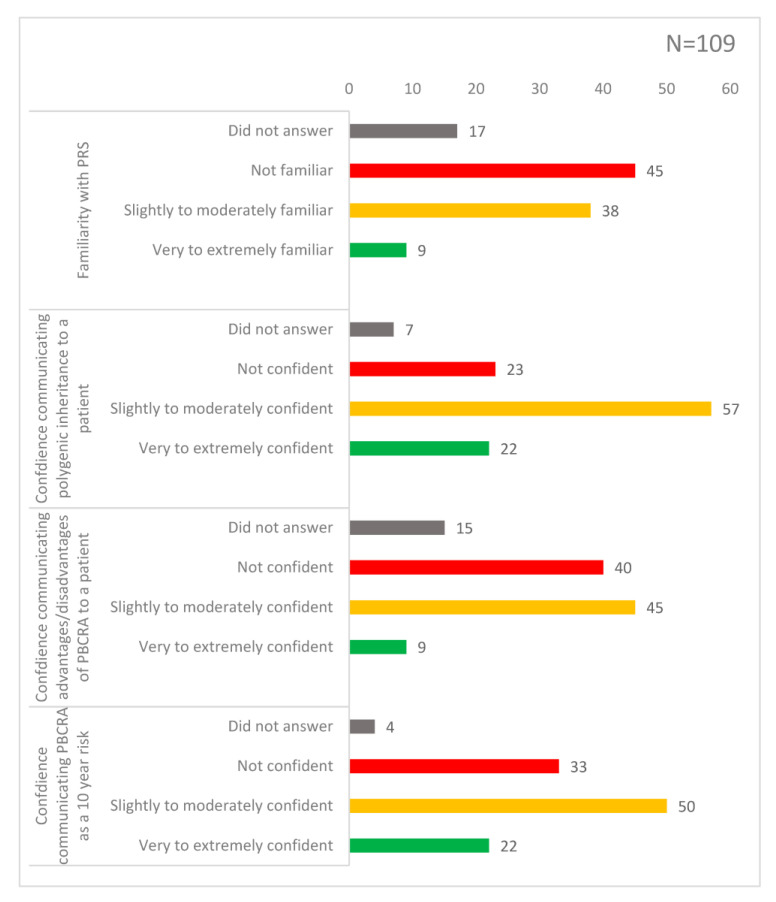
Knowledge of PRS and PBCRA. The bar chart shows responses across four sliding-scale questions assessing familiarity with PRS and confidence communicating concepts and results of a PBCRA to women. Each bar is labelled with the frequency of responses (*n* = 109).

**Figure 2 genes-14-00732-f002:**
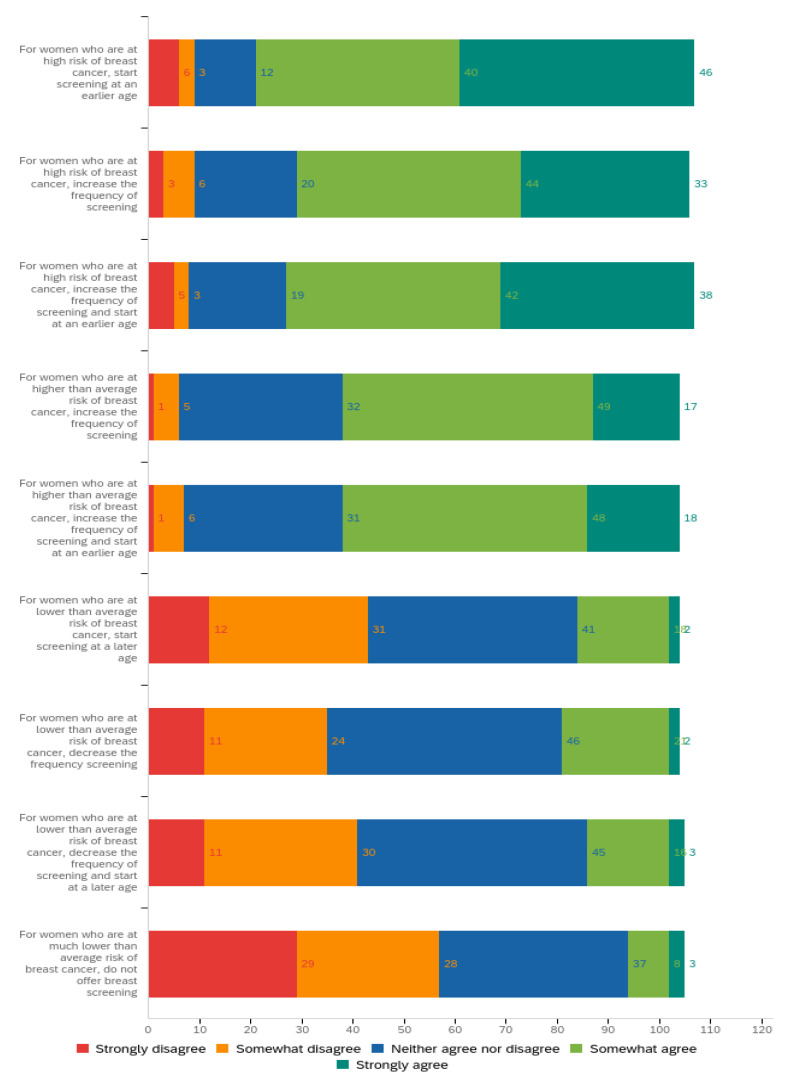
Responses to the statement: ‘A personalised breast cancer risk assessment aims to offer a more targeted screening approach based on individual risk. Please indicate how strongly you agree or disagree with the following recommendations’. Results are displayed as a stacked bar chart labelled with the frequency of responses (*n* = 109) for each option on the Likert scale.

**Figure 3 genes-14-00732-f003:**
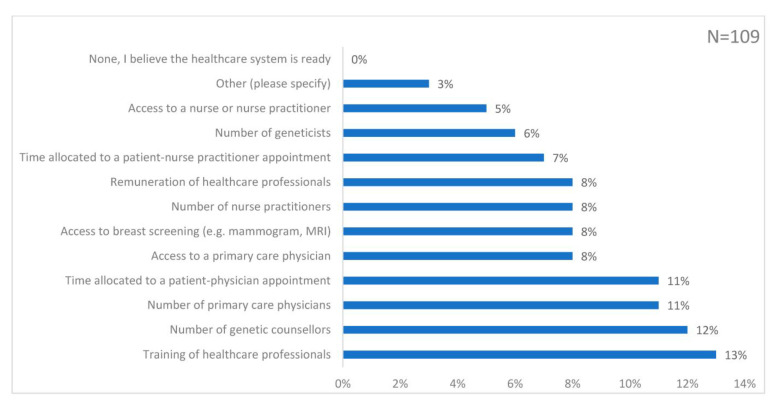
Responses to the question: ‘In your opinion, what aspects of the National Health Service (NHS) should be enhanced to implement breast cancer screening based on a personalised risk assessment?’. The bar chart is labelled with the percentage of responses to each choice.

**Figure 4 genes-14-00732-f004:**
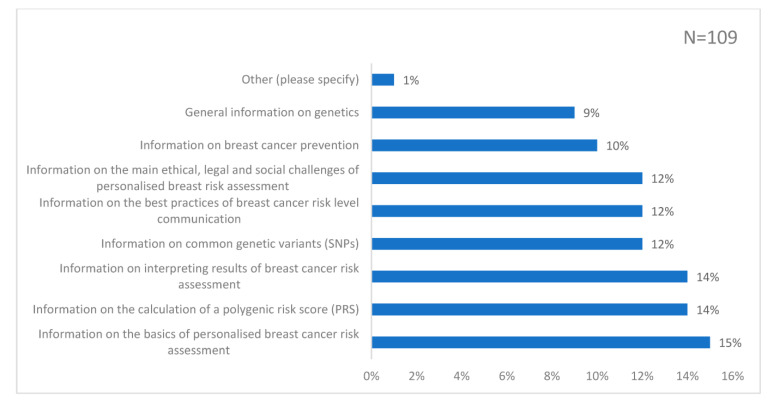
Responses to the question: ‘What type of information would you like to find in the resources you use in your clinical practice to better understand screening based on personalised risk assessment? (Check all that apply)’. The bar chart is labelled with the percentage of responses to each choice.

**Figure 5 genes-14-00732-f005:**
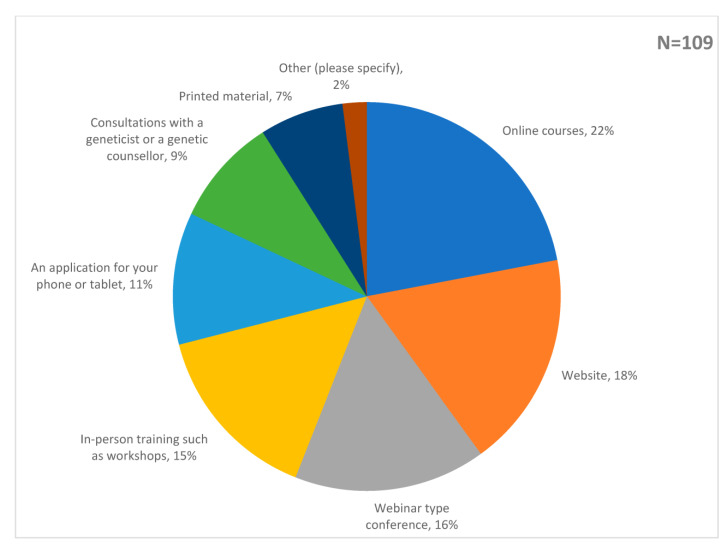
Responses to the statement: ‘For learning more about breast cancer screening based on personalised risk assessment, please select the three resource formats you find most useful’. The pie chart is labelled with the percentage of responses to each choice.

## Data Availability

Not applicable.

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
