# Peer review of "Risk-Stratified Breast Cancer Screening Incorporating a Polygenic Risk Score: A Survey of UK General Practitioners’ Knowledge and Attitudes"

_genes, 2023, doi:10.3390/genes14030732_

Round 1

Reviewer 1 Report

General Comments to the Authors:

This piece looking at attitudes towards personalized risk assessment and polygenic risk scores appears to be seminal for the group considered, General Practitioners. The authors justify their choice of medical professionals receiving the survey and the place of their findings within the currently existent literature. It is always surprising to find out what a third set of eyes can unearth that will strengthen one’s piece. This reviewer recommends the authors go through the suggestions below, one at a time and addressing each in turn, to more fully round out their paper.

Specific Comments to the Authors:

P. 1, line 18:

screening, however,  ->  screening; however,

P. 1, line 19:

unknown._This study aimed to explore GPs’ i) knowledge  ->

unknown. This study aimed to explore GPs’: i) knowledge

P. 1, line 20:

screening ii) attitudes towards risk-stratified screening iii) preferences  ->

screening; ii) attitudes towards risk-stratified screening; and iii) preferences  ->

P. 1, line 21:

development._A cross-sectional  ->  development. A cross-sectional

P. 2, line 49:

Please reiterate the explanation from the Abstract of how a polygenic risk score (PRS) relates to a personalized risk assessment. Also, since you use the term in your Discussion, please also explain how these approaches relate to targeted screening.

P. 2, line 69:

Briefly discuss how the healthcare system in Britain relates to its public health system, and where the fit of GPs lies.

P. 2, line 78:

healthcare systems, then understanding  ->  healthcare systems, understanding

P. 3, line 105:

of a PBRCA  ->  of a [insert full acronym] (PBRCA)

P. 3, line 107:

the current NHSBSP (Likert)  ->  the current [insert full acronym] (NHSBSP) (Likert) 

P. 3, line 109:

screening (MCQ)  ->  screening ([insert full acronym] (MCQ))

P. 3, line 123:

Supply a sentence on what a Bonferroni correction is used for, and another sentence on the effect of such a correction on the statistical critical value.

P. 4, line 145, 3rd Graphic:

advantages/disadvant  ->  advantages/disadvant-

P. 6, line 171, Figure 2. legend:

Supply the total N

P. 7, line 180:

Add a line that these figures contrast with geneticists as a choice focus for NHS attention, and state the % for geneticists.

P. 7, line 187, Figure 3. legend:

Supply the total N

P. 8, line 201, Figure 4. legend:

Supply the total N

P. 8, line 206, Figure 5. legend:

Supply the total N

P. 9, line 212:

no statistically significant associations  ->  no statistically significant associations at the .003 level

P. 9, line 222:

This would be a good point to offer a sentence or two on why no respondents felt the NHS was ready.

P. 9, lines 228-230:

More could be said about the lack of statistical significance. The Bonferroni correction is perfectly acceptable, but it also may be the #1 culprit behind the lack of significance. It would be helpful at this point to state 2 or 3 relations that would have been significant at the P<.05 level if the correction were not undertaken, and reiterate why the correction was necessary. A second reason could have been the closeness of percentages in the alternatives for the given question items. After mentioning these possibilities, would then rephrase your sentence on line 228 to read:

“given these factors, the small sample size made …”

In line 230, you could then speculate that addition of further GP participants could help tease out potential relationships between the questions posed and GP role and years of experience.

P. 9, line 250:

Add a sentence to the effect that PRSs are part of PBCRAs. Also, it would be helpful in this paragraph to restate the two acronyms.

P. 9, line 255:

; and studies  ->  ; studies

P. 9, line 257:

take on this task; this is perhaps  ->  take on the task, which is perhaps

[to avoid using “this” three times in the same sentence]

P. 10, line 277:

Suggest disjoining in this sentence the concept of psychological impact from the concept of too complex and specialist. Instead, speculate on why respondents seem to have missed the opportunity for the NHS to prioritize attention to numbers of geneticists when geneticists are practiced in the use of PRSs.

[this is in regards to your findings in Figure 3.]

P. 10, line 286:

Would mention at this point your thoughts on why printed material was not a highly rated resource format.

[this is in regards to your findings in Figure 5. and in Additional comments]

P. 10, line 292:

Would mention at this point that equipping the NHS workforce includes an appreciation of best practices and referral as well as an understanding of ELSI, which “received an intermediate rating in our study.”

[this is in regards to your findings in Figure 4.]

P. 10, line 284:

key to mitigate  ->  key to mitigating 

P. 10, line 294:

State why population risk-stratification is important to public health.

P. 10, line 310:

To remind your readers, please restate what “BC” and “PRS” stand for.

P. 11, line 315:

Please restate what “NHSBSP” stands for.

P. 11, line 320:

Does “low-risk” in this sentence mean noninvasive, low psychological harm, or offering low-risk results? Please make unambiguous.

Reviewer 2 Report

The manuscript aimed to explore GPs’ knowledge of risk-stratified screening, attitudes towards risk-stratified screening, preferences for continuing professional development.

There are several concerns in the manuscript.

Comments

1. As a basic scientific article, the manuscript has a problem with how to show the Figures and Graphs. The reviewer can not understand the point of making graphs without a horizontal axis. The reviewer really can not understand why the authors chose the bar plot format.

In whole of the manuscript, the authors should remake and revise the figure/graph format.

2. The reviewer concern the main focus of presented manuscript is out of the concept of Genes.

3. Although this manuscript presents several data, the authors' conclusions are not well-founded. Overall, the manuscript is too preliminary to be considered for the publication.

Reviewer 3 Report

I cannot find how may GPs were contacted in total. Only 109 GPs responded to the survey.

You affirm that a minimum of 96 responses were required to achieve a 10% margin of error at a 95% confidence interval (line 88-89). Can you explain how this was calculated?

I would like to suggest to eliminate some of the Figures, ie Figure 2, Figure 4 and Figure 5, the results can be easily described  in the text

Does "96 responses" means that at least  96 partecipating GPs

Round 2

Reviewer 2 Report

The authors did not address to my concerns. The authors don't know how to put the axis on and what is important.

Reviewer 3 Report

Thank you for your answers and revisions